# Structure prediction of protein-ligand complexes from sequence information with Umol

Patrick Bryant ⓘ[1,2,3] ✉, Atharva Kelkar[1], Andrea Guljas[4], Cecilia Clementi ⓘ[4] & Frank Noé ⓘ[1,4,5]

Protein-ligand docking is an established tool in drug discovery and development to narrow down potential therapeutics for experimental testing. However, a high-quality protein structure is required and often the protein is treated as fully or partially rigid. Here we develop an AI system that can predict the fully flexible all-atom structure of protein-ligand complexes directly from sequence information. We find that classical docking methods are still superior, but depend upon having crystal structures of the target protein. In addition to predicting flexible all-atom structures, predicted confidence metrics (plDDT) can be used to select accurate predictions as well as to distinguish between strong and weak binders. The advances presented here suggest that the goal of AI-based drug discovery is one step closer, but there is still a way to go to grasp the complexity of protein-ligand interactions fully. Umol is available at: https://github.com/patrickbryant1/Umol.

Docking of small molecules to protein targets is an important problem for the evaluation of new drugs and the repositioning of known ones[1]. However, existing docking methods have significant limitations: (i) A high-quality structure of the protein is needed as the protein is usually treated at least partially rigid. (ii) The problem of identifying the correct docking pose is not solved[2]. (iii) Most evaluations are performed on structures in their bound (holo) form, limiting the search for new ligands to those which have identical binding modes to known ones[3]. A system that could predict the entire protein-ligand complex structure from a given protein sequence and the chemical structure of the ligand would address these challenges.

Recently, machine learning has been applied to the problem of protein-ligand docking[2,4–6]. However, these systems have not yet outperformed classical methods based on scoring functions[7–9] when considering a known target area or "pocket"[10]. This is a relevant test case as designing new drugs involves targeting specific binding sites on proteins[11], hence one can assume that the binding pocket is known.

On the other hand, it is not reasonable to assume that a protein structure is available in the bound (holo) form consistent with the ligand. When considering structures predicted with ESMfold[12], the success rate (SR, ligand ≤RMSD 2 Å) decreases to half of that on holo-structures (20.3% vs 38.2% of structures) using current state-of-the-art methods[2]. This suggests that pure protein structure prediction tools are not able to produce structures that are suitable for ligand docking.

Evaluation sets partitioned on release date and not on structural similarity is another confounder. When considering receptors dissimilar from those seen during training the SR is about half of that of seen holo receptors (20.8%)[2]. Considering unseen structures and the chemical validity (bond lengths and angles) of the ligands, the SR of some methods can go from 51% to only 1%[13]. Evaluating the same methods on unseen apo (unbound) structures likely results in even lower performance.

Protein flexibility is crucial to reach the holo state and for successful ligand docking. Recently, an all-atom version of RoseTTAFold has been developed. RoseTTAFold All-Atom (RFAA)[14] allows for

[1]Department of Mathematics and Computer Science, Freie Universität Berlin, Arnimallee 12, 14195 Berlin, Germany. [2]The Department of Molecular Biosciences, The Wenner-Gren Institute, Stockholm University, Svante Arrhenius väg 20C, 114 18 Stockholm, Sweden. [3]Science for Life Laboratory, 172 21 Solna, Sweden. [4]Department of Physics, Freie Universität Berlin, Arnimallee 12, 14195 Berlin, Germany. [5]Microsoft Research AI4Science, Karl-Liebknecht Str. 32, 10178 Berlin, Germany. ✉e-mail: patrick.bryant@live.com

predicting proteins in combination with ligands and other biomolecules. The SR on the PoseBusters' test set[13] for protein-ligand prediction is 42%, but it is not known how well the network performs on proteins that have not been seen during training[14]. This suggests that the challenge of protein-ligand prediction is not yet solved.

Here, we develop an AI method that predicts the structure of protein-ligand complexes from sequence information by extending the EvoFormer from AlphaFold2[15]. The network is similar to RFAA, with the difference of not including a 3D track, using template structures or additional crystallographic ligand data as input or during training. In addition, we provide the possibility to specify a binding pocket when this is known as this is often the case in targeted drug development[16].

## Results

Here we develop a protein-ligand co-folding network as a first step towards a Universal molecular framework, Umol (Fig. 1a). Starting from a protein sequence, optional protein target positions (pocket) and ligand SMILES, a multiple sequence alignment (MSA) and a bond matrix are created. From these, features are generated within the network and a 3D structure is produced. There are no limitations on the flexibility of either protein or ligand since no structural information is required to produce the final protein-ligand complex structure.

## Protein-ligand structure prediction

Figure 2a shows the success rate (SR, the fraction of predictions with a Ligand RMSD ≤ 2 Å[17,18]), on 428 diverse protein-ligand complexes[13] for 11 protein-ligand docking methods in addition to Umol. Two different versions of Umol are presented, one that utilises pocket information (Umol-pocket) and one that is completely blind (Umol). Umol, NeuralPlexer1[19] and RoseTTAFold All-Atom (RFAA)[14] are the only methods that do not require native protein structures as input. Umol achieves a SR of 18% and Umol-pocket 45% compared to 24% for NeuralPlexer1 and 42% for RFAA. RFAA without template information (similar structures) amounts to an SR of 8%. The best method is AutoDock Vina[7] with 52% SR but requires both a native holo-protein structure and a target area as input (see Supplementary Table 1 for all SRs).

To see if it is possible to overcome the hurdle of native holo-protein structures, we use AlphaFold2 (AF). Using AF together with DiffDock results in an SR of 21%. The predicted protein pockets have to be highly accurate for successful predictions to be obtained. The average RMSD is 0.91 for the successful models and just slightly above 1 Å (1.23) for the wrong predictions. Since the protein structures are predicted independently of the ligand, it is a priori unclear if a given AF structure is suitable for docking.

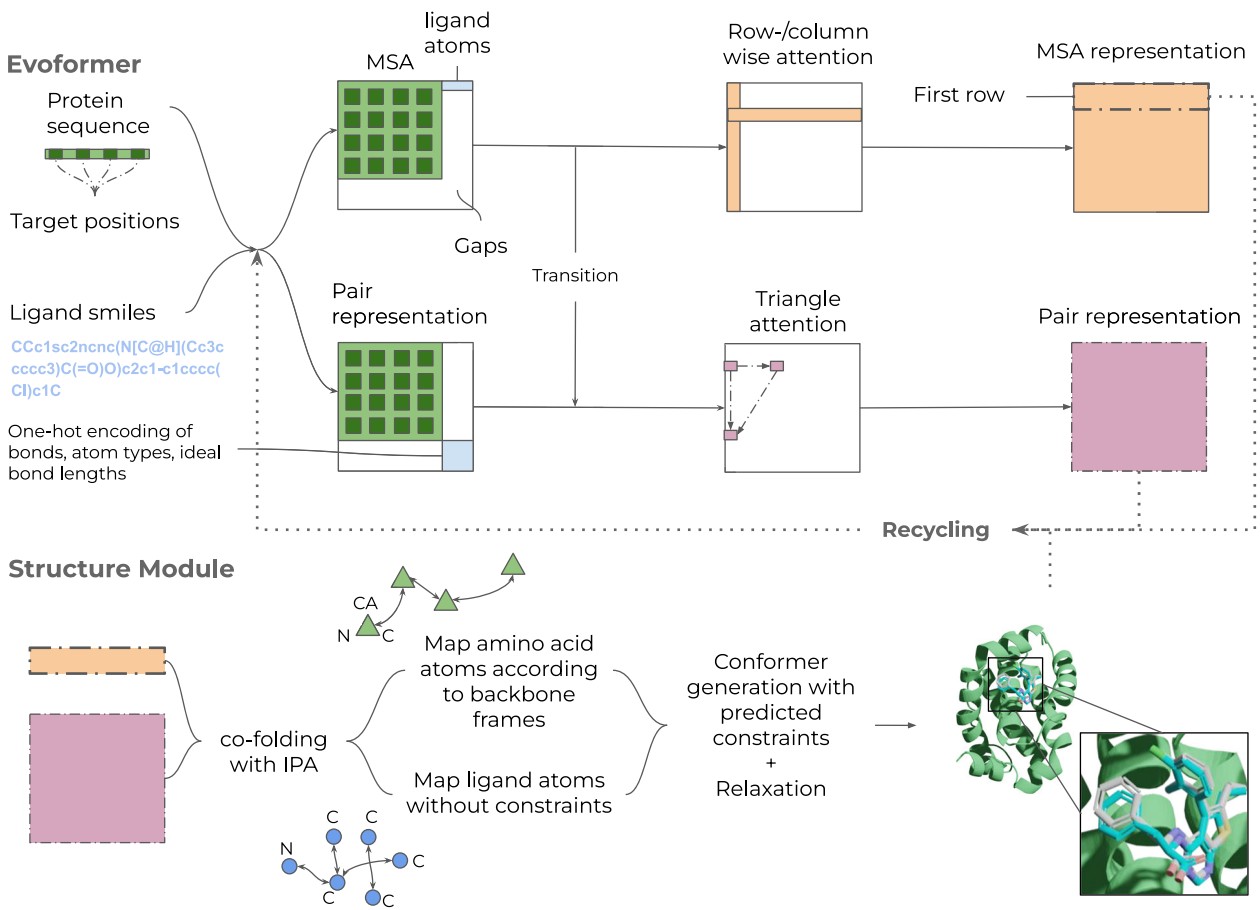

**Fig. 1 | Description of Umol.** An Evoformer network that processes both protein and ligand atom information. There are 48 Evoformer blocks. The protein is represented by a multiple sequence alignment (MSA) and optional target positions in the protein (pocket, Cβs within 10 Å from the ligand) are also defined. When target positions are not known, this information can simply be left out. A SMILES string represents the ligand. Two different tracks are present. The top track processes the MSA (gaps for the ligand) and the bottom track processes pairwise connections within and between the protein and ligand. The resulting representations are fed into the structure module (8 blocks) which produces a 3D structure of the protein-ligand complex. The entire network is trained end-to-end and the representations and predicted atom positions are recycled to refine the final structure. An example for PDB ID 7NB4 (ligand RMSD = 0.57) is shown with the predicted protein in green, the predicted ligand in blue and the native ligand in grey. The ligands are coloured by atomic types (blue=nitrogen, red=oxygen, remaining=carbon). This example was predicted using pocket information.

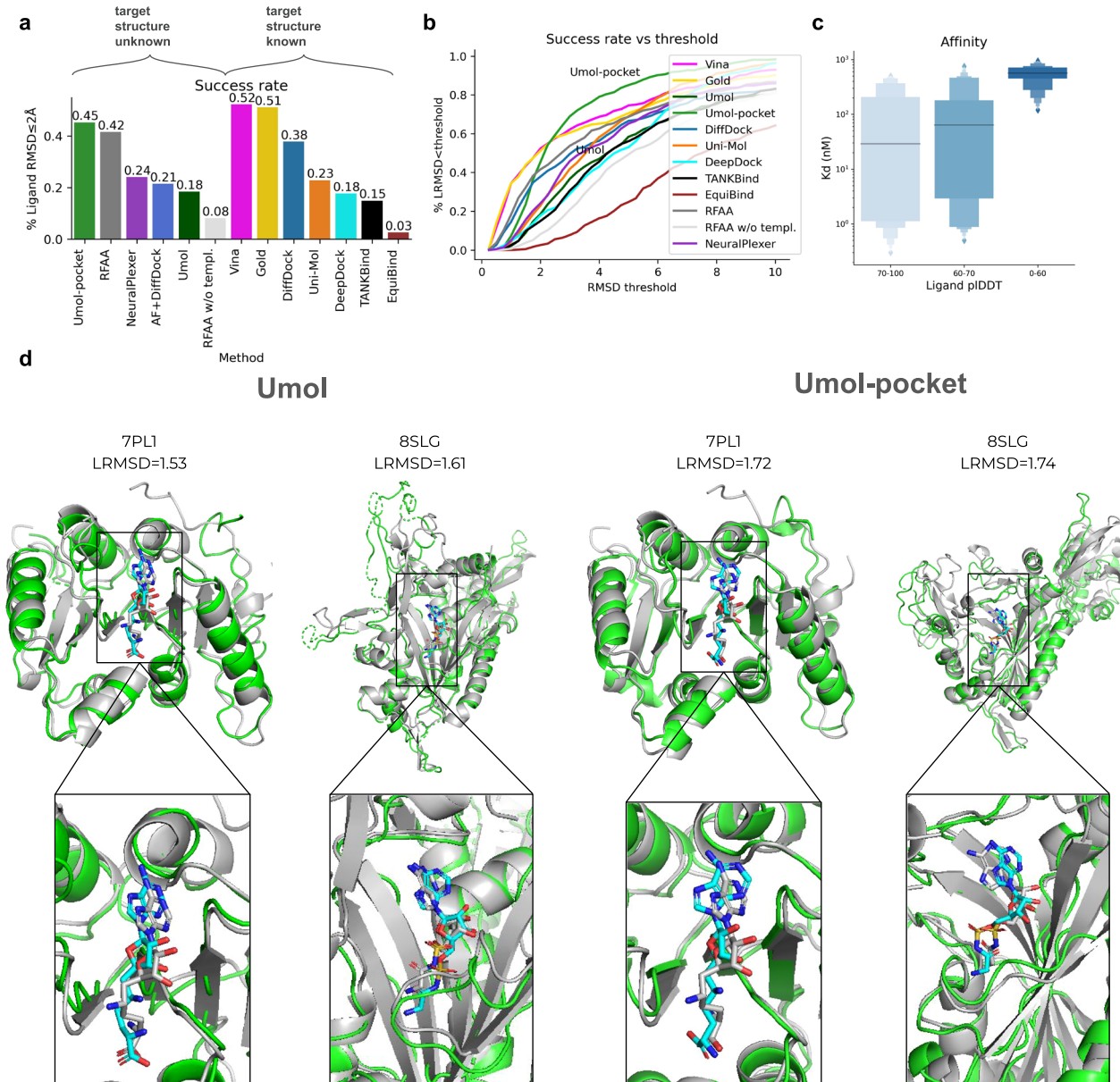

**Fig. 2 | Prediction accuracy. a** Success rates (fraction of predictions with ligand RMSD ≤ 2 Å) on the PoseBuster benchmark set ($n = 428$). Note that only Umol (the method developed here), NeuralPlexer1 and RoseTTAFold All-Atom (RFAA) do not require the native protein structures as input (target structure unknown). The AF +DiffDock predictions are based on structures predicted with AlphaFold2(AF) without ligands. **b** Success rate vs ligand RMSD threshold on the PoseBuster benchmark set ($n = 428$). Umol has many complexes right above the 2 Å mark, suggesting that many ligands are almost in their native configuration. At a threshold of 3 Å, the SR is 69% for Umol vs 58% for Vina. **c** Affinity vs ligand pLDDT for structures predicted with Umol-pocket on a held out test set ($n = 45$). The affinity values (Kd) are from experimental studies extracted from the PDB. Values

with a maximum of 1000 nM (1 mM) were selected. There is only one point above 80 and one below 50 plDDT, which is why the thresholds differ from that in 2c. At >70 plDDT, the median affinity (horizontal lines) is 30 nM, while it is >500 nM at <60 plDDT. The centre boxes encompass data quartiles and horizontal lines mark the medians for each distribution with min/max values marked by diamonds. **d** Examples of predictions from the Posebusters test set with low homology to the Umol training set (<30% sequence identity). The native structures are in grey and the predicted protein/ligand in green/cyan. The ligands are coloured by atomic types (blue = nitrogen, orange=phosphorus, red = oxygen, remaining=carbon). PDB IDS: 7PL1 and 8SLG with ligand RMSDs 1.53, 1.61 vs 1.72, 1.74 for Umol and Umol-pocket respectively.

A success cutoff of 2 Å ligand RMSD (LRMSD) is arbitrary. Figure 2b shows the SR vs the ligand RMSD threshold. Umol-pocket has many complexes right above the 2 Å mark, suggesting that many ligands are almost in their native configuration. At just above 2 Å (2.35 Å), Umol-pocket surpasses all other methods and at a threshold of 3 Å, the SR is 69% for Umol vs 58% for Vina. Umol-pocket has no successful complexes below 0.5 Å, but Vina and Gold do. This is likely a consequence of these methods using the native structures as an input, resulting in errors of close to 0 Å which should not be possible in a realistic setting.

A highly important aspect of drug design is affinity[20,21]. Figure 2e displays the affinity (Kd) for 45 held-out targets without homology to the Umol training set vs. the Umol-pocket ligand plDDT (see below). At above 70 ligand plDDT, the median affinity is 30 nM, while it is above 500 nM below 60 plDDT. This suggests that high- and low-affinity targets can be distinguished based on the ligand plDDT as well as the accuracy of the ligand positions (Fig. 2c). At really high affinity (<10 nM, $n = 13$), there is a strong correlation with the ligand plDDT (Pearson R = −0.77). Affinity can

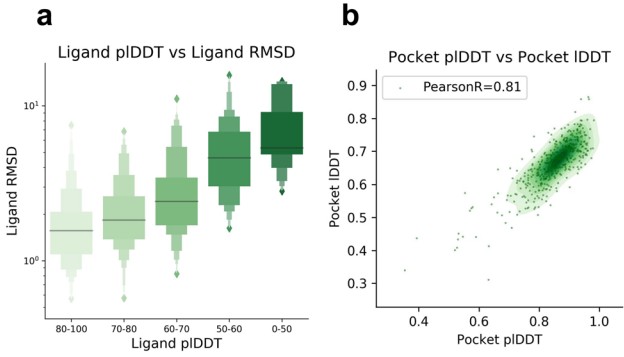

**Fig. 3 | Confidence metrics and accuracy. a** Ligand plDDT vs ligand RMSD for structures predicted with Umol-pocket (n = 428). The success rate (ligand RMSD ≤ 2 Å) for each bin is 80-100: 72.3%, 70-80: 58.2%, 60-70: 34.4%, 50-60: 5.3%, 0-50: 0.0%. The centre boxes encompass data quartiles and horizontal lines mark the medians for each distribution with min/max values marked by diamonds. **b** Protein pocket (all CBs within 10 Å from any ligand atom) predicted lDDT (plDDT) vs pocket lDDT for structures predicted with Umol-pocket (n = 428) as a density plot with each datapoint represented as a green scatter. The Pearson correlation coefficient is 0.81, suggesting a strong relationship between the two. **c** Ligand

plDDT vs ligand RMSD for structures predicted with Umol (n = 428). The centre boxes encompass data quartiles and horizontal lines mark the medians for each distribution with min/max values marked by diamonds. The success rate (ligand RMSD ≤ 2 Å) for each bin is 80–100: 50.0%, 70–80: 52.6%, 60–70: 16.7%, 50–60: 1.8%, 0–50: 1.2%. At a plDDT threshold of >85, the SR is 80% for Umol without pocket information, enabling the selection of blind predictions with high confidence. **d** Protein pocket predicted lDDT (plDDT) vs pocket lDDT for structures predicted with Umol (n = 428) as a density plot with each datapoint represented as a green scatter. The Pearson correlation coefficient is 0.78.

also be distinguished without pocket information (see section BindingDB).

Figure 2d shows examples of predictions from both Umol and Umol-pocket with low homology to the training set (<30% sequence identity) in structural superposition with the native complexes. Umol-pocket predicts the protein structures very well, but not all protein regions are entirely accurate for Umol. The position of the ligand relative to the protein interface is accurate with both Umol and Umol-pocket, suggesting that both methods can be used to obtain accurate predictions.

### Confidence metrics and chemical validity

To see if accurate predictions can be distinguished from inaccurate ones based on the Umol model outputs, we analyse the relationship between the ligand RMSD and the predicted lDDT[22] (plDDT, Fig. 3). At plDDT >80, the SR is 72% and <50 plDDT 0% with Umol-pocket suggesting that accurate ligand poses can be distinguished (Fig. 2a, b). The same is true for Umol without pocket information (Fig. 2c, d) where the SR is 80% >85 and 1.2% <50. The protein pocket (all CBs within 10 Å from any ligand atom) plDDT displays Pearson correlations of 0.81 and 0.78 with the lDDT for Umol-pocket and Umol, respectively.

Previous AI methods produce ligands that are not chemically valid[13]. Since we use RDKit, the generated ligands will make sense chemically. 98% of ligands are valid for Umol-pocket according to PoseBuster's ligand criteria. We also conclude that the proteins are predicted with high accuracy overall (Supplementary Fig. 1), reporting an average TM-score[23] of 0.96 with Umol-pocket.

### Structure prediction of BindingDB

BindingDB contains experimentally determined protein-ligand binding affinity measurements curated from literature (https://pubmed.ncbi.nlm.nih.gov/17145705/).

29583 of the measurements are below 1000 nM in either KI, IC50, KD or EC50 and have less than 600 residues. We predicted the structure of these with Umol (no pocket information) to investigate what advancements in structural annotation can be expected.

Figure 4 shows affinity metrics vs the ligand plDDT for 27810 complexes that were successfully predicted. All experimental techniques show a relationship with the ligand plDDT, suggesting that Umol can distinguish between low- and high-affinity binders to some degree even without pocket information. The biggest separation is for Kd measurements where the median affinity is 10 nM at plDDT values > 80

and 80 nM <50 plDDT. The lowest separation is for EC50, which increases up to 50 plDDT where it drops suddenly. This is reflected in the ROC AUC scores for selecting high-affinity binders (<20 nM, KD AUC = 0.66 and p-value = 1.58e-17, EC50 AUC = 0.55 and p-value = 0.059, Supplementary information). We note that previous studies have pointed out issues with affinity data compiled from different studies[24]. At an FPR of ≈10%, ≈45% of high-affinity KD binders can be accurately selected (Supplementary Fig. 5).

We selected predicted complexes that have a ligand plDDT>85 as these are 80% likely to be correct (Fig. 2) resulting in 336 complexes from 42 different proteins. Figure 4e shows examples from different classes of proteins with both single ligands and clusters of different ligands together as predicted in the same structures. We note that even though these structures seem plausible and their L-plDDT scores are high, they remain experimentally verified.

## Discussion

We have introduced Umol, a neural network for predicting the complete all-atom structure of protein-ligand complexes only from protein sequence information, the position of the binding site in the sequence (optionally), and the chemical graph of the ligand. Umol does not depend on any structural information in contrast to all other ligand docking methods that rely on native holo protein structures or template information. Compared to its closest relatives, RoseTTAFold All-Atom and NeuralPlexer1, Umol obtains a higher success rate (SR, Ligand RMSD ≤ 2 Å) when including pocket information on the PoseBusters test set (45% vs 42% and 24%, respectively) making it the highest performing method for protein-ligand structure prediction.

When pocket information is removed from Umol and template information from RFAA, the SR drops to 18% and 8%, respectively. When using DiffDock with AF predictions, the accuracy is 21% but is dependent on highly accurate interface predictions (pocket RMSD < 1 Å). All methods except for RFAA have higher performance on structures similar to those in the training set, suggesting potential data leakage in the training or validation procedure for RFAA (Supplementary Table 1).

Many ligand poses slightly above the success threshold of 2 Å are likely equivalent, which indicates that a more flexible scoring system may be required. This is exemplified by the fact that Umol surpasses the success rate of AutoDock Vina at a threshold of 2.35 Å. In cases where the native protein structures are not used for the scoring, even small errors in alignment become an issue.

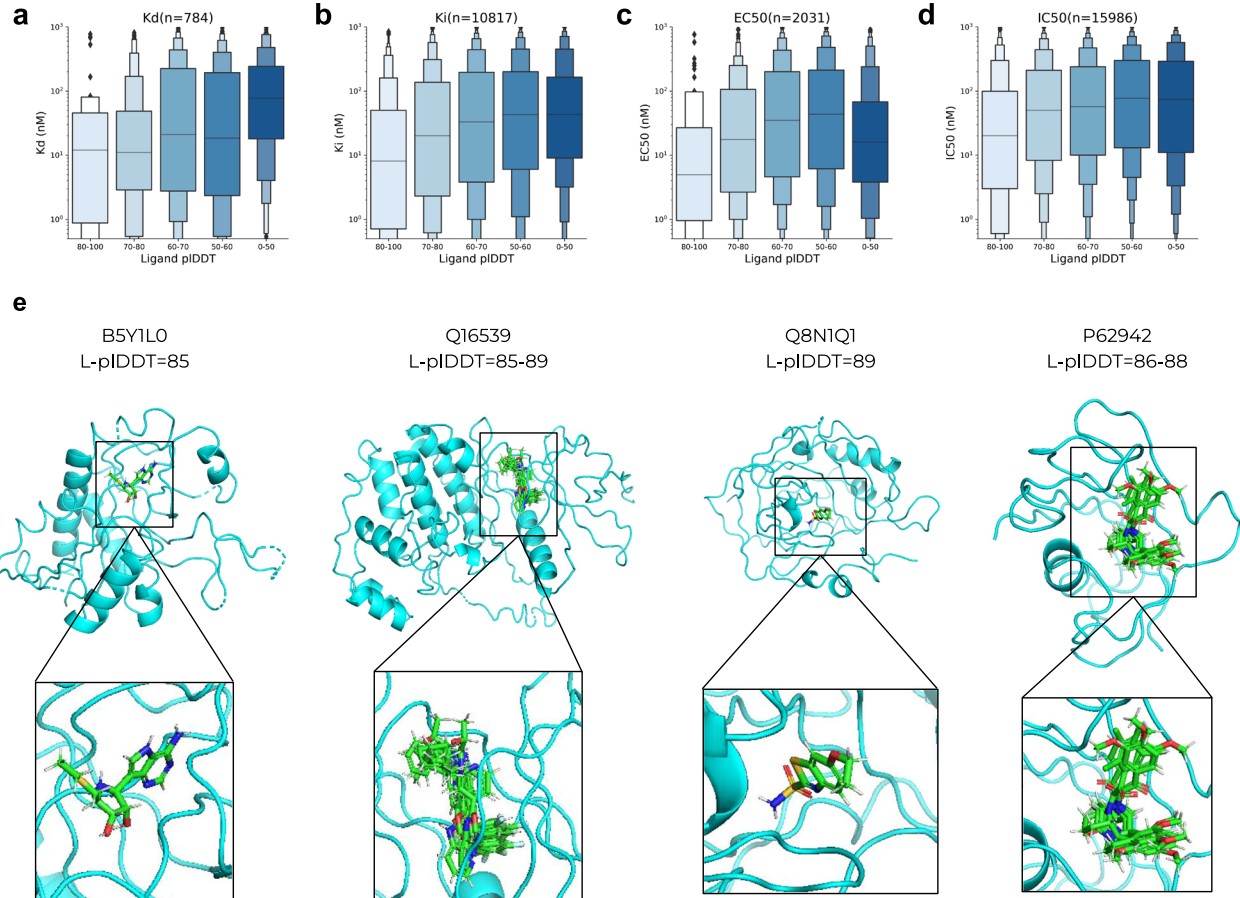

**Fig. 4 | BindingDB predictions. a–d** Affinity (log scale) KD, Ki, EC50 and IC50, respectively vs average Umol ligand plDDT binned in steps of 10 (no pocket information, n = 27810). The centre boxes encompass data quartiles and horizontal lines mark the medians for each distribution with min/max values marked by diamonds. All experimental measures show a relationship with the ligand plDDT suggesting that the network can distinguish strong from weak binders. We also calculate the *p*-values between the affinity distributions using a selection of plDDT <50 or >80. The Corresponding *p*-values (one-sided t-test associating having a higher affinity value with a lower plDDT) are 1.58e-17, 5.45e-18, 0.0052 and 0.059 for Kd, Ki, EC50 and IC50 data, respectively (**a–d**). **e** Examples of predicted protein-ligand complexes from BindingDB. The UniProt IDs and the range of ligand plDDT (L-plDDT) scores are displayed above the structures. We note that even though these structures seem plausible and their L-plDDT scores are high, they remain to be experimentally verified.

Co-folding protein-ligand complexes has the potential to accelerate drug repositioning. In particular, we find that the predicted lDDT (plDDT) of the ligand can be used to select accurate docking poses and that the plDDT of the protein pocket is suitable for selecting accurate interfaces (Fig. 3). The ligand plDDT also separates high- and low-affinity ligands, suggesting that some of the predictions that Umol and Umol-pocket are uncertain about may be weak binders (Figs. 2 and 4). This further speaks for the ability of Umol and underlines that important aspects of protein-ligand interactions seem to have been learned.

Although the accuracy without pocket information is 18%, the network can still separate strong and weak binders to a certain degree. This is especially useful for annotating unknown complexes and we present 336 protein-ligand structures with very high confidence (ligand plDDT>85). We note that even though these structures seem plausible and their L-plDDT scores are high, they remain to be experimentally verified.

We do not find clear relationships between the model's predictive performance and different features related to the protein or ligand (Supplementary Fig. 2). However, we do find that among the cases where other methods struggle, Umol-pocket is accurate in 3 out of 5 cases (Supplementary Fig. 3). By inverting the trained network, it is possible that new ligand-binding proteins or protein-binding ligands can be designed. Another option is using transfer learning to create a

generative diffusion model for the same purposes[14,25]. In such settings, the ligand or protein plDDT could be maximised to attempt the creation of high-affinity binders.

The current release of PDBbind contains data processed from PDB in 2019. Since then, many more protein-ligand complexes have been submitted suggesting that higher accuracy may be possible to achieve. However, it is unclear what accuracy is needed to obtain meaningful results for protein-ligand docking. The high accuracy observed in protein structure prediction is not obtained for tasks involving other molecules, such as small molecules or RNA[26–28]. Without coevolutionary information on proteins, the accuracy of the structure prediction decreases rapidly[12,15]. As there is no similar source of information for small molecules or RNA, one has to rely solely on atomistic representations.

Here, we suggest that pocket information is useful, as we observe that the Deep Learning methods seem prone to overfitting without it (Supplementary Table 1). This finding is further exemplified by the observation that although many of the molecules in the PoseBusters test set contain highly similar analogues in the training dataset (calculated by the Tanimoto similarity coefficient, Supplementary Fig. 4) this similarity does not correlate with model success.

Overfitting is not observed to the same extent for structure-based docking methods like Vina or Gold (Supplementary Table 1). This is expected as these are based on atomistic scoring functions and

**a**

Length distribution

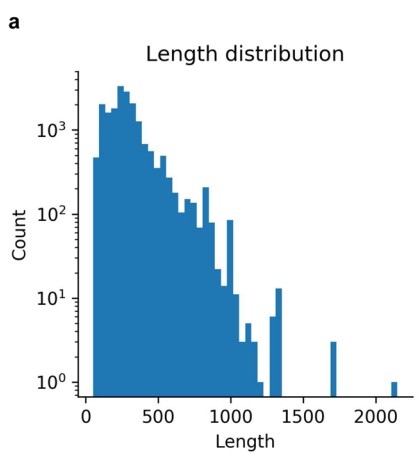

**b**

Cluster size distribution

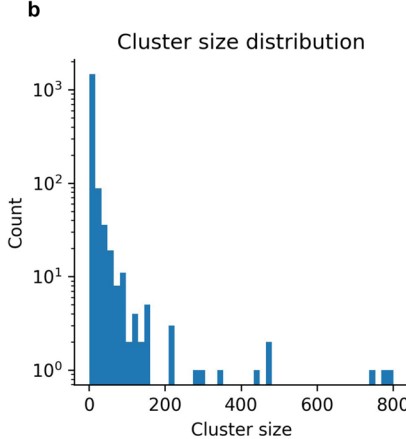

**Fig. 5 | Sequence metrics. a** Length distribution (n = 17936, median length=263 residues). **b** Number of sequences in each 20% sequence identity cluster (n = 1652, median cluster size = 2).

thereby do not rely on protein homology to the same extent. The issue with Deep Learning methods having a substantially higher performance on the training set suggests that protein homology plays a significant role in protein-ligand docking. We note here that RFAA has higher performance on the test set than the training set, suggesting potential data leakage between the train and test sets.

There is still a long way to go to grasp the complexity of protein-ligand interactions fully, but leveraging deep learning for structure prediction of the entire complex may bring us one step closer to the solution.

## Methods
### PDBbind
We used PDBbind from 2019 (2020 release[29]) processed by the authors from EquiBind (https://zenodo.org/record/6408497, 19119 protein-ligand complexes). We parsed all protein sequences from the PDB files. 18884 out of 19119 protein structures (99%) could be parsed (<80% missing CAs and >50 residues). Only the first protein chain in all protein-ligand complexes used here and in the evaluation was extracted. Features (see below) could be generated for 17936/18884 (95%) protein-ligand complexes. The failed ones did so due to issues of converting SMILES to 3D structures using RDKit (version 2023.03.2, https://www.rdkit.org).

### Data partitioning
We cluster all protein sequences with MMseqs2 (version f5f780acd64482cd59b46eae0a107f763cd17b4d)[30] at 20% sequence identity with the options:

```
mmseqs easy-cluster DB.fasta clusterRes tmp --min-seq-id
0.2 -c 0.8 --cov-mode 1
```

We obtain 1486 sequence clusters at 20% sequence identity, representing the number of possible protein folds. Most sequences are

below 1000 residues (median=263 residues, Fig. 5a) and most clusters have only a few entries (Fig. 5b). We select 90% of the clusters for training, 5% for validation and 5% for eventual calibration tasks (used for the affinity analysis) (Table 1).

Note that the main evaluation and comparison with other methods (Fig. 2, Supplementary Table 1) is performed on the PoseBusters benchmark set that only includes structures not included in PDBbind 2020 (described below). The partition of the data in PDBBind was made before the PoseBusters benchmark was available, whereupon a decision to use that test set instead was made.

### Affinity data from the PDB
We extracted affinity values from the PDB using the calibration set. In total 516 out of 671 examples have some kind of affinity measure, but these vary widely in technology and accuracy. Therefore, we chose to focus on only Kd values in the high accuracy range (<1000 nM), resulting in 45 protein-ligand complexes with affinity values of which 13 have affinities <10 nM. None of the protein-ligand complexes in the PoseBusters test set has available affinity data in the PDB, which is why this set was not used.

### BindingDB
There are 92366 affinity measurements curated from literature (2024-12-18) in Binding DB (https://pubmed.ncbi.nlm.nih.gov/17145705/). 44890 have some affinity measure (['Ki (nM)', 'IC50 (nM)', 'Kd (nM)', 'EC50 (nM)']) that is below 1000 nM and most (60%) are <100 nM. The proteins consist of 1000 unique UniProt IDs and there are 27993 unique ligands. We limit the protein length to 600 residues, resulting in 29583 complexes from 705 UniProt IDs and 18530 ligands, 62% of these have affinity <100 nM. 27810 (94%) of the complexes were successfully predicted. The ones that failed did so due to MSA generation errors resulting in missing alignments.

### PoseBusters test set
We evaluate the trained network and compare it with 8 other methods for protein-ligand docking on the PoseBusters benchmark set which contains 428 complexes not present in the PDBbind 2020 release[29] used for training here. We also compared the performance by dividing the dataset based on the sequence overlap to the training set (Supplementary Table 1)[13].

We downloaded the PoseBusters set and scores for all methods except for RFAA from https://zenodo.org/record/8278563. The success rate for RFAA was taken from their preprint[14].

We predicted all test complexes using three recycles (as for the validation set) and only one prediction was run per target. 15

**Table 1 | Data partitioning for training, validation and calibration (affinity)**

| Partition | Number of clusters | Number of protein-ligand complexes |
|---|---|---|
| Train | 1486 | 16420 |
| Valid | 82 | 845 |
| Calibration (affinity) | 84 | 671 |
| Total | 1652 | 17936 |

complexes are out of memory using an NVIDIA A100 GPU with 40 Gb of RAM during inference (above 1000 residues). We crop these to 500 residues to include as many of the target residues (pocket) as possible.

In the published version of the PoseBusters study, any proteins that include crystal contacts were removed from the evaluation[13]. We decided to include the set from the original study as a method for predicting protein-ligand structures should not be influenced by experimental artefacts. The effect of the crystal contacts on the protein-ligand interaction can also not be known and many other experimental artefacts such as different conditions for protein expression, cofactors and ligand concentrations exist.

### Network description and inference
The network architecture is a modification of the EvoFormer used in AlphaFold2[15]. For an overview of the network, see Fig. 1. Note that no template information is used in the network, making it purely sequence-based. Here, we outline the encoding and processing of the different network inputs.

### Pocket encoding
All CB (CA for Glycine) within 10 Å from any ligand atom are one hot encoded. This is an arbitrary threshold similar to that of DeepDock[31]. Likely, similar thresholds will perform equally well. The encoding is used to bias both the MSA and pair representations and is processed throughout the network.

### No pocket version
For the version of Umol that does not use pocket information, the one hot encoding is simply omitted. Otherwise, the networks are identical allowing for inference with the same code but different sets of network weights.

### Ligand encoding
All atoms present at least 100 times in the ligands of the training dataset (B, C, F, I, N, O, P, S, Br, Cl) are one hot encoded. All other atoms are encoded with the same hot encoding representing rare ligand atoms (As, Co, Fe, Mg, Pt, Rh, Ru, Se, Si, Te, V, Zn). We decided not to encode rare atoms to not overfit to these and to save space as this would result in a much sparser atom encoding.

The ligand bonds are one hot encoded as well. We encode single, double, triple and aromatic bonds separately and all other bonds as rare.

### MSA generation and processing
To generate multiple sequence alignments, we search uniclust30_2018_08[32] with HHblits (from HH-suite[33] version 3.1.0) with the options:

```
hhblits -E 0.001 -all -oa3m -n 2
```

We add a gap for the ligand atoms in the MSA representation and process the MSA in the MSA track alone. The MSA track is aware of the ligand through interactions with the pair track and due to the size and encoding of this in the initial MSA representation.

### Pair processing
We process the pair representation with the atoms directly: amino acids+atoms, and let the MSA information flow to the pairwise interactions to influence the folding.

### Recycling operations
The pair representation, the first row of the MSA representation and intermediate predicted final atom positions are recycled through the network 1-3 times sampled uniformly during training. For the predictions, three recycles were used.

### Loss
We use the same main loss functions and loss weights as in AlphaFold2, but with slight modifications to better suit the problem of protein-ligand structure prediction. The loss used up to step 24500 is

$$Loss = 0.5 \cdot FAPE + 0.5 \cdot AUX + 0.3 \cdot Distance + 0.2 \cdot MSA + 0.01 \cdot Confidence \tag{1}$$

Where $FAPE$ is the frame aligned point error, $AUX$ a combination of the $FAPE$ and angular losses, $Distance$ a pairwise distance loss, $MSA$ a loss over predicting masked out MSA positions and $Confidence$ the difference between true and predicted lDDT scores. These losses are defined exactly as in AlphaFold2 and we refer to the description there[15].

The FAPE is calculated using only the amino acid N-CA-C frames towards ligand atoms as well. The MSA loss is only applied for the protein as well, but the distance and aux loss include both protein and ligand.

All ligand atoms are represented as their own frames (like a gas of atoms without constraints). The alternative is to enforce bond distances and known geometric properties. To enforce accurate bond lengths, we introduce an L2 distance loss at step 24500. This L2 distance loss is defined using the distance from the ideal ligand bond lengths extracted from a generated conformer in RDKit (clipped at 10 Å). The reason for not including this initially is to let the network learn freely how the protein and ligand should interact. This loss has a weight of 0.1 and is added into the AUX loss. This loss acts as a form of harmonic potential for the ligand bonds.

### Training and validation
We sample the sequences with inverse probability to the 20% sequence identity cluster size. We use a learning rate of 0.001 with 1000 steps of linear warmup and clip the gradients with a global norm of 0.1 as in AlphaFold2[15]. The optimiser is Adam[34] applied through the Optax package in JAX (JAX version 0.4.23, https://github.com/deepmind/jax/tree/main). We train the pocket network for 50000 steps and the no-pocket network for 90000 steps (until convergence) across 8 NVIDIA A100 GPUs, with a combined batch size of 24 (3 examples per GPU).

We crop the protein-ligand complexes to 256, where the protein size is 256 subtracted with the ligand size (number of atoms, the entire ligand is always included) residues. The protein is cropped uniformly around the protein pocket region. Each step takes approximately 18 seconds. We validate every 10000 steps and assess the ligand RMSD and the protein pocket RMSD using the relaxation and scoring procedures described below.

### Training with pocket information
The loss function declines rapidly (Eq. 1, Fig. 6a). The masked MSA loss saturates quickly (Fig. 6b), while the distogram loss is noisy throughout the training procedure (Fig. 6c). The structure module loss (Fig. 6d) is not saturated at step 50,000, although the lDDT[22] for the protein and ligand only improves slowly (Fig. 6e). Figure 6f suggests that the network starts to overfit to the training set between steps 40,000–50,000. The success rate on the validation set is the highest at step 40,000 (26%). The total training time is approximately 10 days.

Both the lDDT of the protein and ligand are improving together, but there is a tradeoff in the SR. The pocket RMSD is the highest at the step where the SR is (Fig. 6g and h), suggesting a tradeoff between these two. There is a difference of only 0.18 Å in the median pocket RMSD between the validation checkpoints, but the SR differs 8.6% (Fig. 6h).

### Training without pocket information
The training without pocket information closely follows that of when pocket information is available (Fig. 7). The SR is much lower without pocket information though, suggesting that this is needed to obtain

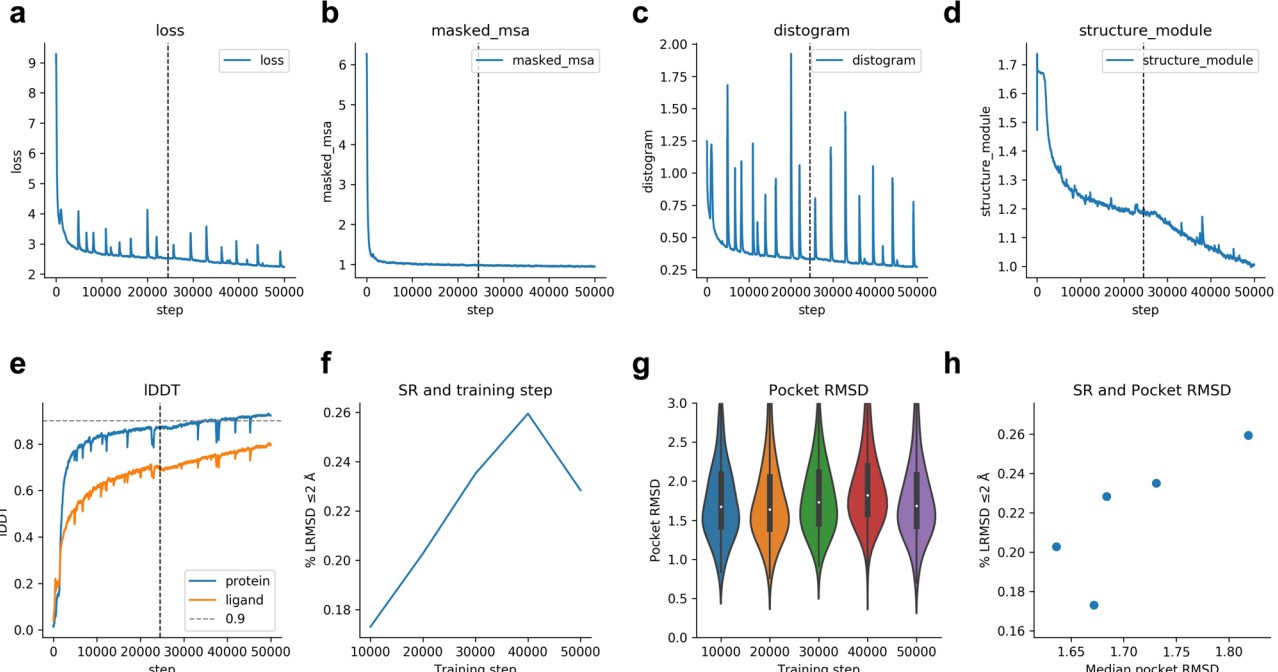

**Fig. 6 | Training curves and metrics for Umol-pocket.** The dashed vertical line indicates the time point where the additional L2 ligand distance loss was introduced (step 24500). **a** The combined loss function (Eq. 1) vs the training step. **b** Masked MSA loss vs training step. **c** Distance loss vs training step. **d** Structure module loss vs training step. This is the combination of FAPE and AUX (Eq. 1) **e** lDDT for protein and ligand vs training step. The protein accuracy is higher than that of the ligand. **f** Success rate (SR, % of protein-ligand complexes predicted with ligand RMSD ≤ 2 Å) for the validation set (n = 741 protein-ligand complexes) at checkpoint intervals of 10000 steps. The examples that failed (n = 104) did so due to RAM limitations and missing interface atoms. The SR is 17.3, 20.3, 23.5, 25.9

and 22.8 for steps 10000-50000, respectively. **g** RMSD of the atoms in the protein pocket for the validation set (n = 741 protein-ligand complexes) at checkpoint intervals of 10000 steps. The examples that failed (n = 104) did so due to RAM limitations and missing interface atoms. Boxes encompass data quartiles, horizontal lines mark the medians and upper and lower whiskers indicate respectively maximum and minimum values for each distribution. **h** Median pocket RMSD vs SR for the validation set (n = 741 protein-ligand complexes) at the different checkpoints. The examples that failed (n = 104) did so due to RAM limitations and missing interface atoms.

more accurate results. Interestingly, the lDDT training curves are very similar between the two runs. Since the lDDT curves are similar, resulting in a similar training set performance, but with the validation SR being 20% lower. This suggests that the network overfits to certain proteins when pocket information is not present.

The total training time is approximately 19 days. The highest validation SR is obtained at 60000 steps and this checkpoint was used for all analyses here.

## Conformer generation with RDKit
Conformer generation with RDKit is important to ensure that the predicted structures have realistic bond angles[13] and to adjust for small errors in e.g. the predicted protein side chain angles. Since the ligand atoms are represented as a point cloud, there are no constraints on bonds or angles resulting in possible violations in the predictions. To fix these issues, we align conformers generated by RDKit (version 2023.03.2, https://www.rdkit.org) from the input SMILES string using the ligand atom distance matrix as constraints to the predicted structures. We generate a total of 100 conformers and select the one with the lowest difference in atomic positions to the predicted positions.

## Relaxation with openMM
We noticed that some of the predictions contain clashes (here defined as two atoms being less than 1 Å apart). This is a common problem with protein structure prediction[15], but can be easily amended using fast energy minimisation in a molecular dynamics force field, so-called relaxation. To relax the predicted structures, we add hydrogens to the

protein and ligand and minimise the energy using OpenMM (version 8.0)[35]. We constrain the protein CA positions and the ligand positions, meaning that only the protein side chains are moved significantly.

The energy minimisation does not improve the ligand RMSD but fixes clashes and other errors within the protein. We use a Brownian Integrator and minimise the energy of the protein with a tolerance of 1 kJ for a maximum of 1000 steps with a restraining force of 100 kJ/nm². This is a very fast relaxation procedure that essentially only alters side chain positions.

For the Posebuster test set, the relaxation resolves clashes in the protein-ligand interfaces. Before relaxation, 12% of interfaces predicted with Umol-pocket have at least one clash, compared to 0.7% after the relaxation. The success rate is diminished by 2%, resulting in an SR of 43.3% compared to 45.3% before relaxation. This is a marginal change and we allow the user to determine if this is acceptable or not through the choice of relaxing the predicted structures or keeping them in their unrelaxed states.

## Inference
For inference, the same information as generated for training is used. An MSA is generated for the protein (see above) and the ligand atoms are encoded directly (see above). The difference between Umol-pocket and Umol is a simple one-hot encoding of what residues have CBs within 10 Å from any ligand atom. In all other aspects, Umol and Umol-pocket are identical in their architectures.

## Timings
Table 2.

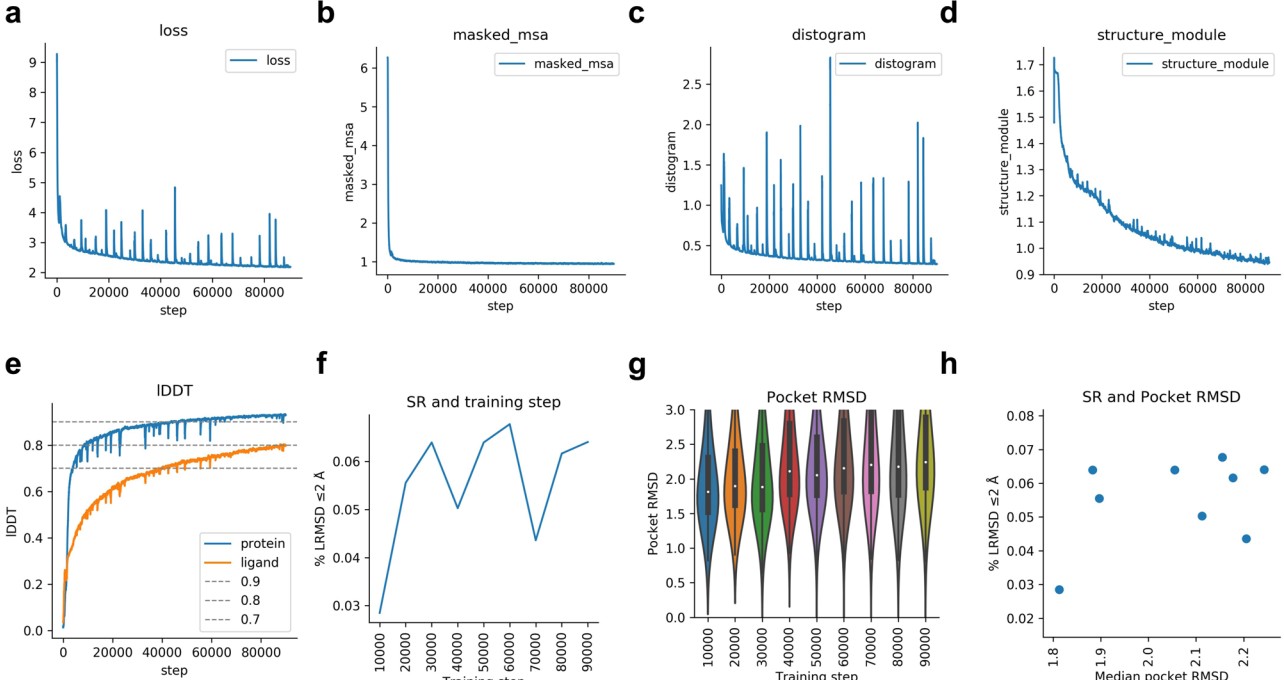

**Fig. 7 | Training curves and metrics for Umol. a** The combined loss function (Eq. 1) vs the training step. **b** Masked MSA loss vs training step. **c** Distance loss vs training step. **d** Structure module loss vs training step. This is the combination of FAPE and AUX (Eq. 1) **e** lDDT for protein and ligand vs training step. The protein accuracy is higher than that of the ligand. **f** Success rate (SR, % of protein-ligand complexes predicted with ligand RMSD ≤ 2 Å) for the validation set (n = 741 protein-ligand complexes) at checkpoint intervals of 10000 steps. The examples that failed (n = 104) did so due to RAM limitations and missing interface atoms. The SR is 2.8, 5.6, 6.4, 5.0, 6.4, 6.8, 4.4, 6.2 and 6.4 for steps 10000-90000, respectively. **g** RMSD of the atoms in the protein pocket for the validation set (n = 741 protein-ligand complexes) at checkpoint intervals of 10000 steps. The examples that failed (n = 104) did so due to RAM limitations and missing interface atoms. Boxes encompass data quartiles, horizontal lines mark the medians and upper and lower whiskers indicate respectively maximum and minimum values for each distribution. **h** Median pocket RMSD vs SR for the validation set (n = 741 protein-ligand complexes) at the different checkpoints. The examples that failed (n = 104) did so due to RAM limitations and missing interface atoms.

**Table 2 | Average time per process (the "real" time from the bash script is reported) needed for generating features and predicting the protein-ligand complex structures**

| What | Average time | Computational resources |
|---|---|---|
| MSA generation | 285 s | 8 ×2.5 GHz Intel(R) Xeon(R) CPU E5-2680 v3 with 2.5 Gb RAM |
| Network input feature generation | 11 s | 2 ×2.5 GHz Intel(R) Xeon(R) CPU E5-2680 v3 with 2.5 Gb RAM |
| Protein-ligand structure prediction | 182 s | 1 x NVIDIA A100 GPU with 40 Gb RAM |
| RDKit ligand conformer generation | 13 s | 1 ×2.5 GHz Intel(R) Xeon(R) CPU E5-2680 v3 with 2.5 Gb RAM |
| OpenMM protein relaxation | 27 s | 1 x NVIDIA RTX A4000 with 16 Gb of RAM |
| Total | 518 s ≈ 9 minutes | At least 2 ×2.5 GHz Intel(R) Xeon(R) CPU E5-2680 v3 with 2.5 Gb RAM and 1 x NVIDIA A100 GPU with 40 Gb RAM |

The most time-consuming process is the MSA generation and the least is the network feature generation. The time is the same for Umol and Umol-pocket.

## Comparison methods

We compare the performance on the PoseBusters benchmark[13] with the methods evaluated there for protein-ligand docking and the recently released RoseTTAFold All-Atom (RFAA). We describe the different methods briefly below:

**AutoDock Vina.** A classical docking method based on perturbing the ligand structure and calculating a score until convergence. Requires the native holo protein structure, a description of possible ligand poses, and a cube with a side length of 25 Å centred on the centre of mass of the heavy ligand atoms[7].

**Gold.** A classical docking method based on perturbing the ligand structure and calculating a score until convergence. Requires the native holo protein structure, a description of possible ligand poses, and a cube with a radius of 25 Å centred on the centre of mass of the heavy ligand atoms[36].

**RoseTTAFold All-Atom.** RoseTTAFold All-Atom (RFAA) is a neural network for predicting interactions between proteins and ligands as well as other atoms such as metal ions. RFAA lets each ligand atom move freely and treats the amino acids as N-CA-C frames as in Umol. The biggest differences to Umol are the "3D track", the input of template and sterical information, the treatment of each ligand atom as an individual frame in the loss calculations (FAPE). Compared to Umol-pocket the biggest difference is the lack of defining a pocket.

In Umol, the FAPE is only calculated as seen through the amino acid frames. This is because even though all ligand atoms can have frames, they will be highly variable depending on the predicted positions of the other ligand atoms. In contrast, the amino acid frames will be constant as the N-CA-C relationship remains the same regardless of the predicted positions.

As in Umol, RFAA uses relaxation of the predicted protein-ligand structures. RFAA uses Rosetta[37], while Umol uses molecular dynamics with OpenMM. Another difference in the loss calculations is that RFAA

reorders all ligand atoms to find the lowest possible loss. The reason for not applying this here is that it is very difficult to exactly determine what atoms are interchangeable. Often, atoms can be interchangeable on a local scale, but not when considering the whole ligand[14].

**RoseTTAFold All-Atom w/o templates.** To analyse the impact of known structures on the outcome, we reran RFAA on the PoseBusters test set without templates using the same MSAs as for Umol and their default configuration. We find that the SR decreased substantially from that reported using templates (8% vs 42%[14], Supplementary Table 1). The protein structures are predicted with high accuracy (average protein pocket RMSD = 1.43), suggesting that the coevolutionary information for the protein is sufficient. The most reasonable explanation for the decreased performance is, therefore, the exclusion of interface information available through protein structural templates.

**DiffDock.** DiffDock takes ligand SMILES and the native holo protein structure as input and generates amino acid embeddings with ESM2. An initial ligand conformation is generated with RDKit which is updated and docked to the native protein structure using a diffusion process and deep learning. DiffDock does not require a defined pocket[2].

**AF+DiffDock.** We predicted the protein structures with AF v2.1, model 1 (parameters available from https://storage.googleapis.com/alphafold/alphafold_params_2021-07-14.tar) using 3 recycles and 1 ensemble[15]. We then input these structures to DiffDock[2] and generated one model per target structure.

**NeuralPlexer version 1.** NeuralPlexer1 uses a diffusion concept on amino acid frames and ligand atoms. We ran NeuralPlexer1 with the default config "batched_structure_sampling" (16 samples, 40 steps and a chunk size of 4), langevin_simulated_annealing and only protein sequence information for the protein as input. This ensures that the protein-ligand complex is co-folded. We select the first-ranked models for scoring[19].

**Uni-Mol.** A deep learning method that takes a ligand and native holo protein structure as input. Requires all protein residues within 6 Å of any heavy ligand atom[38].

**DeepDock.** A deep learning method that takes a ligand and native holo protein structure as input. Requires a protein surface mesh of all protein residues within 10 Å of any heavy ligand atom[31].

**TankBind.** A deep learning method that takes a ligand and native holo protein structure as input. Does not require a defined pocket[4].

**Equibind.** An SE(3)-equivariant geometric deep learning method that takes a ligand and native holo protein structure as input. Does not require a defined pocket[5].

### Scoring protein and ligand poses
To score the predicted protein-ligand complexes, we align the pocket CAs to the native structures and transform the predicted ligand accordingly. We then calculate the ligand RMSD and the RMSD of all available atoms in the protein pocket (Eq. 2). We assume that the atomic order is the same for the predicted/native ligands and do not adjust for symmetry. This can result in a lower success rate in some cases, but very few cases should be affected by symmetrical swaps beyond the 2 Å RMSD threshold.

$$RMSD = \frac{1}{n}\sum_{i=1}^{n}\sqrt{\sum_i (pc_i - nc_i)^2} \qquad (2)$$

Where $pc$ is the predicted 3D atomic coordinate $(x, y, z)$ and $nc$ is the native 3D atomic coordinate, and n is the number of heavy (non-hydrogen) atoms.

We score the overall protein structures with TM-align (version 20220412)[39] and the command:

```
TMalign native.pdb predicted.pdb
```

### Number of effective sequences (Neff)
We clustered sequences at 62% identity to estimate the amount of information available in each MSA of the test set[40]. The resulting number of clusters were used as the number of effective sequences (Neff). We used MMseqs2 (version f5f780acd64482cd59b46eae0a107 f763cd17b4d)[30] with the command:

```
mmseqs easy-cluster MSA clusterRes tmp --min-seq-id 0.62
-c 0.8 --cov-mode 1
```

### Reporting summary
Further information on research design is available in the Nature Portfolio Reporting Summary linked to this article.

## Data availability
The predicted structures used for the calculation of various metrics presented in the figures here as well as the input features for the training of the network and summary statistics used to produce all figures can be found at: https://zenodo.org/records/10809161.

## Code availability
Umol is available at: https://github.com/patrickbryant1/Umol.

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

## Acknowledgements

This study was supported by the European Commission (ERC CoG 772230 "ScaleCell", F.N.), MATH+ excellence cluster (AA1-6, AA1-10, F.N), Deutsche Forschungsgemeinschaft. (SFB 1114 porjects A04 and C03, RTG 2433 Project Q04, SFB/TRR 186 Project A12, F.N), the BMBF (Berlin Institute for Learning and Data, BIFOLD, F.N), the Einstein Foundation Berlin (Project 0420815101, C.C) and SciLifeLab & Wallenberg Data Driven Life Science Program (grant: KAW 2020.0239, P.B). Computational resources were obtained from ZIH (SCADS) at TU Dresden with project id p_scads_protein_na (P.B and F.N) and from LiU with project ids Berzelius-2023-267 and Berzelius-2024-78 (P.B). We thank the authors of PoseBusters for their extensive benchmark and for making the data available. We also thank the OpenMM team for discussions regarding the structure relaxation.

## Author contributions

P.B. designed and performed the studies, prepared the figures and wrote the initial draft of the manuscript. A.K. performed the relaxation of the predicted structures in consultation with P.B and A.G. A.G. extracted affinity values from PDB, calculated Tanimoto similarity and analysed the relationship with plDDT in consultation with A.K. and P.B. All authors contributed in reading and improving the manuscript draft. C.C. and F.N. obtained funding. P.B. and F.N. obtained computational resources.

## Funding

## Competing interests

The authors declare no competing interests.
