## [Peer Review File · Nature Communications]

Reviewers' Comments:

Reviewer #1:

Remarks to the Author:

Umol is a method for the co-folding of protein with a ligand. The method extends AlphaFold2 architecture to ligand atoms similarly to the recently published RosettaFold-All atoms.

In the revised version authors have added comparisons to additional methods, including NeuralPlexer and RosettaFoldAA. Umol has a performance comparable to other methods. Overall, the paper shows that the problem is challenging and far from being solved.

ROC curves were added to show that ligand pLDDT vs. affinity. I find it strange that only pLDDT values below 50 and above 80 were used, the result is a single-point ROC curve which doesn't say much about significance. The whole idea of the ROC curve is not to use any cut-offs. I recommend adding a p-value comparing the relevant boxplots in Figs. 4a-d

Reviewer #2:

Remarks to the Author:

I have reviewed the manuscript previously for Nature Method. I would like to thank the authors for addressing most of my concerns.

My remaining concern involves overfitting. The author mentioned in their response the potential overfitting of RF-AA. I'm curious if Umol, too, might be affected by this issue. I would suggest adding a discussion about overfitting of these ML methods to guide the readers about when to apply these techniques and when to use classical approaches.

We are delighted that the manuscript is deemed suitable for publication and thank the reviewers and editors for their time and valuable insights. We have taken the final adjustments into consideration and marked the corresponding changes in red in the manuscript.

REVIEWERS' COMMENTS

Reviewer #1 (Remarks to the Author):

Umol is a method for the co-folding of protein with a ligand. The method extends AlphaFold2 architecture to ligand atoms similarly to the recently published RosettaFold-All atoms.

In the revised version authors have added comparisons to additional methods, including NeuralPlexer and RosettaFoldAA. Umol has a performance comparable to other methods. Overall, the paper shows that the problem is challenging and far from being solved.

ROC curves were added to show that ligand pIDDT vs. affinity. I find it strange that only pIDDT values below 50 and above 80 were used, the result is a single-point ROC curve which doesn't say much about significance. The whole idea of the ROC curve is not to use any cut-offs. I recommend adding a p-value comparing the relevant boxplots in Figs. 4a-d

We thank the reviewer for this suggestion and have added p-values to the relevant boxplots in Figure 4 and to the supplementary information (supplementary figure 5):

We also calculate the p-values between the affinity distributions using a selection of pIDDT <50 or >80. The Corresponding p-values (one-sided ttest associating having a higher affinity value with a lower pIDDT) are 1.58e-17, 5.45e-18, 0.0052 and 0.059 for Kd, Ki, EC50 and IC50 data, respectively (a-d).

Reviewer #2 (Remarks to the Author):

I have reviewed the manuscript previously for Nature Method. I would like to thank the authors for addressing most of my concerns.

My remaining concern involves overfitting. The author mentioned in their response the potential overfitting of RF-AA. I'm curious if Umol, too, might be affected by this issue. I would suggest adding a discussion about overfitting of these ML methods to guide the readers about when to apply these techniques and when to use classical approaches.

We thank the reviewer for this suggestion and have added a paragraph in the discussion about this issue:

Overfitting is not observed to the same extent for structure-based docking methods like Vina or Gold (Supplementary Table 1). This is expected as these are based on atomistic scoring functions and thereby do not rely on protein homology to the same extent. The issue with Deep Learning methods having a substantially higher performance on the training set suggests that protein homology plays a significant role in protein-ligand docking. We note here that RFAA has higher performance on the test set than the training set, suggesting potential data leakage between the train and test sets.